# Real-Time Integrative Mapping of the Phenology and Climatic Suitability for the Spotted Lanternfly, *Lycorma delicatula*

**DOI:** 10.3390/insects16080790

**Published:** 2025-07-31

**Authors:** Brittany S. Barker, Jules Beyer, Leonard Coop

**Affiliations:** 1Oregon Integrated Pest Management Center, Oregon State University, 2215 Cordley Hall, Corvallis, OR 97331, USA; beyerju@oregonstate.edu (J.B.); coopl@oregonstate.edu (L.C.); 2Department of Horticulture, Oregon State University, 4017 Agriculture and Life Sciences Building, Corvallis, OR 97331, USA

**Keywords:** invasive pest, developmental model, forecast, thermal stresses, establishment risk, decision support, degree-day

## Abstract

The spotted lanternfly (SLF), *Lycorma delicatula* (White, 1845) (Hemiptera: Fulgoridae), is expected to cause major disruptions to fruit, ornamental, and forest industries in the United States and other countries. We present a model that predicts the seasonal development (phenology) of the SLF in areas that are climatically suitable for establishment. The model was developed using previously published data and presence records from the native range. An independent set of presence records and phenological observations was used for the model validation. The model performed well but needs further validation using monitoring data from across the SLF’s distribution in North America. Cold and heat stresses may be insufficient to exclude the SLF in most parts of the conterminous United States; however, an inability for the pest to complete its life cycle in cold areas may hinder establishment. Feeding pressure on host plants may be greater in warm regions owing to the earlier appearance of adults. The near-real-time forecasts produced by the model are available at two websites to support decision making for the surveillance and management of the SLF.

## 1. Introduction

Native to northern China [1], the spotted lanternfly (SLF), *Lycorma delicatula* (White, 1845) (Hemiptera: Fulgoridae), is an invasive planthopper that was first detected in North America in 2014 in Berks County, Pennsylvania, and has since spread to at least 18 states in the eastern United States [2,3,4]. This highly polyphagous pest feeds across a wide range of economically important plants, including grapes, peaches, apples, and walnuts, as well as plants in natural and residential areas [5,6,7]. Additionally, the SLF’s production of honeydew may indirectly impact plants by attracting secondary pests and causing sooty mold growth, which impairs photosynthesis and potentially results in plant dieback or death [3,8,9]. The economic damage caused by the SLF is significant, with an expected annual loss of USD 195.2 million to Pennsylvania’s agriculture and forestry industries alone [3,10].

Reducing the threats posed by the SLF to agricultural and natural resources depends on conducting surveillance, monitoring, and management of the pest in the right place at the right time of the year [11]. Focusing surveillance in areas at high risk of establishment may increase the likelihood of detecting and responding to the SLF early, before it spreads to additional locations. Establishment risk models for the SLF include both process-based models and correlative climatic suitability models at regional and worldwide scales [12,13,14,15,16,17,18]. However, most models use averages of historical climates, which provide only a single snapshot of establishment risk based on temperatures that were cooler than present-day climates owing to global warming [19,20,21]. Additionally, they do not capture the potential impacts of extreme weather events, such as hard freezes, that may reduce the SLF’s survival [22,23,24].

In areas that are suitable for establishment, forecasts of the SLF’s phenology may aid surveillance because detection efforts often target the nymph and adult stages, which are more observable than egg masses [25]. Additionally, phenological forecasts may help with the timing of insecticide and entomopathogen treatments, some of which are more effective when applied at a particular point in the pest’s life cycle [26,27,28,29]. Published phenology models developed for the SLF include those that predict the dates of spring egg hatch and adult emergence [30,31,32] as well as the pest’s monthly development and mortality [14]. A phenology model for the SLF, developed for the Spatial Analytic Framework for Advanced Risk Information Systems (https://safaris.cipm.info/, accessed on 1 March 2025) [32], produces near-real-time forecasts of the emergence of nymphs and adults for the conterminous United States (CONUS) to aid in surveillance operations. However, none of these models simulates developmental variation within populations, which is known to occur in the SLF [24], or integrates phenological forecasts with estimates of the establishment risk.

In this study, we present a model that integrates the mapping of the phenology and climatic suitability for the SLF for use in the Degree-Day, Risk, and Pest Event Maps (DDRP) platform. DDRP was designed to provide decision support for detecting, monitoring, and managing invasive pests [33]. The DDRP model for the SLF improves upon previously developed models for this pest by integrating predictions of phenology and climate-based establishment risk, predicting phenology for all the major life stages, allowing for within-population variation in development, and having the ability to predict climatic suitability using current and forecast climate data. We evaluated the model using observations from monitoring studies and biodiversity databases. Forecasts of egg hatch and the appearance of adults for North America and Europe were produced to demonstrate the model. Near-real-time forecasts for the CONUS are available at two web sites to support early detection efforts and management programs for the SLF.

## 2. Methods

All the analyses and modeling were conducted using R (v. 4.1) [34]. Raster data processing and calculations were performed using the “terra” R package (v. 1.7) [35]. The plots were produced using the “ggplot2” R package (v. 3.4.0) [36].

### 2.1. Modeling Overview

The DDRP platform assumes long-established norms for phenological modeling used in decision support, such as making use of linear degree-day models that use daily minimum and maximum temperatures (*T_min_* and *T_max_*, respectively) as model inputs. Details on the mechanics and data requirements of DDRP already exist [33,37] and are, therefore, only summarized for this study. We believe that the more advanced use of non-linear equations is not warranted in the case of this model because (1) temperature–development studies and data used to parameterize the SLF’s development rates [30,38] were well-fitted using the linear approach (see Section 2.3.2), (2) a non-linear modeling approach may not offer significant improvement in model accuracy in general [39] and would require temporal resolutions (hourly or better) that are not currently available from high-quality spatial data sources, and (3) spatial models, such as DDRP, would be challenged by computational limits if non-linear developmental submodels and higher-resolution temperature data were to be incorporated. Therefore, we adhered to the simpler, widely used standards in the use of linear degree-days in DDRP.

### 2.2. Climate Data

The modeling used daily estimates of *T_min_* and *T_max_* for the most-recent 20-year timeframe available for each focal region. These included the Community Data Analysis Tools (CDAT) dataset for China from 1999 to 2018 at a spatial resolution of 0.1° [40] (https://zenodo.org/record/5513811#.YtnLNXbMIuU, accessed on 28 June 2022) and the Daymet dataset for North America from 2004 to 2023 at a spatial resolution of 1 km^2^ [41,42] (https://daymet.ornl.gov/getdata, accessed on 11 September 2024). The Daymet data were cropped to a maximum latitude of 60° N and aggregated to a spatial resolution of 4 km^2^ to decrease model run times.

### 2.3. Phenology Model Development

#### 2.3.1. Life Cycle and Overwintering Stage

The phenology model uses a starting date of 1 January and assumes a one-year (univoltine) life cycle in which the egg is the overwintering stage. The model stops after one complete life cycle (obligate_diapause = 1) because of evidence that eggs require a chilling period before development can resume [43,44]. Four life stages (egg, larval, pupal, and adult) and a separately parametrized overwintering stage are included in DDRP models. To conform to these naming conventions for the SLF, we defined the nymph stage (instars 1–4) as the “larva”, the pre-oviposition period as the “pupa”, and the first-to-50%-oviposition period as the “adult”.

#### 2.3.2. Thresholds, Stage Durations, and Phenological Events

We reinterpreted temperature vs. development rate data for the SLF from several laboratory studies [30,38] and monitoring studies [5,30,45,46,47,48]. For lab studies conducted at multiple temperatures, we used linear regression (the x-intercept method) with forcing through the x-intercept to estimate the lower temperature threshold and degree-day requirements for the major life stages. From these data, we solved for a common lower temperature threshold of 10 °C for eggs, larvae, pupae, and egg-to-adult. The results all had high R-squared values, ranging from a low of 0.964 (egg) to a high of 0.988 (instar 4), with a combined (instars 1–4) value of 0.996. Common thresholds for life stages are required by the site-based (weather station) phenology modeling platform at USPest.org (https://uspest.org/risk/models, accessed on 29 July 2025). Additionally, using the same thresholds for both platforms facilitates the cross-comparison of predictions.

The stage durations for the major life stages derived using regression equations are reported in Table 1. At a lower threshold of 10 °C, the egg, nymphal, egg-to-adult, and pre-oviposition degree-day requirements were 202, 890, 1092, and 630 degree-days, respectively. The first-to-50% oviposition was estimated as 146 degree-days. The model calculates degree-days using the single triangle calculation method with an upper threshold [49]. It predicts five phenological events: egg hatch, nymphs halfway developed, adult emergence, oviposition, and diapausing egg.

The model applied seven time-distributed cohorts to approximate a normal distribution (distro_shape = “normal”) of hatch times in overwintered eggs (Table 1). Variability in egg development has been observed in laboratory studies that reared eggs at constant temperatures [24,43] and in monitoring studies, which documented an overall normal distribution in the counts of eggs and nymphs during a season [48,50,51]. Egg-monitoring data from five sites in Pennsylvania and Virginia, USA [5,30,45,47,48,51], were used to calibrate values for the minimum (xdist1), mean (distro_mean)*,* and maximum (xdist2) degree-days to the complete development. For each site-year (N = 15), we extracted and analyzed predictions of the minimum (earliest) and average (peak) dates of egg hatch that were produced by models with different parameter combinations. This included estimating the number of overpredictions (too late) as well as the mean absolute error (MAE) and bias (in days) (Appendix A). All the models applied 15,000 degree-days for the variance of the developmental completion times (distro_var). The parameter combination that resulted in the fewest number of overpredictions was used in the final model (Appendix A).

#### 2.3.3. Phenology Model Validation

The first dataset used to validate the phenology model consisted of monitoring data derived from the literature (N = 20) and the Nature’s Notebook database (N = 6) from 17 unique locations between 2017 and 2020 in the eastern United States (Figure 1A, Appendix A) [23,52,53,54]. Nature’s Notebook data were downloaded using the “rnpn” R package (v. 1.25) [55,56]. We estimated the differences between the predicted and observed DOYs (DOY_pred_ and DOY_obs_, respectively) of the first dates of egg hatch, nymphs halfway developed, appearance of adults, and oviposition. DDRP models do not include larval instars; therefore, the DOY_pred_ for the “nymphs halfway developed” event was compared to the DOY_obs_ of the late nymphs (instars 3 and 4). A linear regression model with 95% confidence intervals and 95% prediction intervals was used to evaluate the level of the precision of predicted dates and the uncertainty around individual observations, respectively. Additionally, we calculated descriptive statistics for each event, including the MAE and bias.

The second dataset used to validate the phenology model included georeferenced observations of eggs, nymphs, and adults from the United States and China, derived from the iNaturalist database (Figure 1B). Observations were downloaded using the “rinat” R package (v. 0.1.9) [57] and filtered to retain only those with a “research-grade” status for years in which model predictions were available (2014−2023 for the United States and before 2018 for China). To retain independence among data points, we reduced multiple observations of a life stage for the same date and location (resulting from an observer submitting multiple photographs) to a single observation. Egg observations for the specified time range were only available for the United States.

Photographs for each iNaturalist observation were inspected by a single researcher (J.B.) to verify that the correct species and life stage were documented. Additionally, each nymph observation was categorized as either an early nymph (black with white spots) or a late nymph (red with white spots and black stripes) based on their coloration. Egg observation dates exhibited a clear bimodal distribution that was consistent with documented dates of egg hatch (typically by late June) and oviposition (beginning in September) in the northeastern United States (Appendix A) [8,48,58]. We therefore categorized egg observations as overwintered vs. first-generation eggs based on the date (before 31 June vs. after 1 September, respectively). Egg observations collected in July and August were excluded because the corresponding photos indicated that hatching had already occurred. This resulted in 12,982 georeferenced observations for the United States (N = 12,927) and China (N = 55). Table 2 summarizes the observations for each country.

The iNaturalist observations were single encounters with the SLF at specific dates; therefore, we only used them to assess whether a life stage was predicted earlier than it was observed because its date of first appearance was unknown. We estimated the difference (in days) between DOY_pred_ and DOY_obs_ for each iNaturalist observation and determined which life stage DDRP predicted in relation to the observed stage. The differences were visualized with histograms, and a one-sided paired Wilcoxon signed-rank test was conducted in R (“wilcox.test”) to assess whether the DOY_pred_ of the first appearance of a life stage was significantly earlier than DOY_obs_ (i.e., DOY_pred_ < DOY_obs_). For the overwintered eggs, however, we assessed whether the latest date of egg hatch occurred after DOY_obs_ (i.e., DOY_pred_ > DOY_obs_) because egg hatch is the first event predicted by the model. A non-parametric test was chosen owing to evidence for non-normality in the data.

### 2.4. Climatic Suitability Model Development

In DDRP, areas where a species is not excluded by survival-limiting temperature stresses are considered to be a part of the potential distribution and, therefore, at the highest risk of establishment [33]. Stress units accumulate when temperatures exceed the cold- and heat stress thresholds and exclude the species if they exceed the moderate or severe limits. Moderate stress may inhibit long-term establishment, in which short-term (one complete year) establishment may occur only during favorable years, whereas areas under severe stress would likely prevent even short-term establishment. Areas excluded by moderate stress may also be used to communicate uncertainty in the potential for establishment [33].

We developed the climatic suitability model for the SLF, using eco-physiological information and presence records from China, and then evaluated its performance, using records from the United States. The records were obtained from the literature, theses, reports, the Global Biodiversity Information Facility [59] (GBIF, http://gbif.org, accessed on 1 November 2024), and the Early Detection and Distribution Mapping System (EDDMapS, https://www.eddmaps.org, accessed on 20 November 2024) [60]. We excluded GBIF records with a geographic uncertainty of >10 km and those from regions with no documented establishments of the SLF. Additionally, we removed records that were not collected during the 20-year timeframe used for modeling and those with missing climate data. This filtering process resulted in 14,987 records for the United States and 145 records for China.

#### 2.4.1. Cold Stress Parameters

Estimates of *T_min_* of the coldest week were extracted for each presence record from China to help to identify a cold stress threshold. We averaged and then aggregated daily CDAT data to a weekly resolution to reflect longer-term (i.e., multi-day) cold stress experienced by the SLF [22,50,61]. According to this analysis, 98% (385/392) of the records occurred in areas with average weekly *T_min_* values of ≥−16 °C (Appendix A). This result is consistent with a monitoring study that found very low egg survival rates (<2%) at South Korean sites where the average daily *T_min_* fell below −16 °C [50], as well as with experiments that documented very low hatch rates in eggs that experienced −15 °C for 10 and 15 days [62].

The severe cold stress limit corresponded with areas of China that experienced more than ca. 14 consecutive days below −16 °C (Appendix A). In a laboratory study, the survival rates of cold-acclimated SLF eggs dropped from ca. 5% to 0% after exposure to −15 °C for seven and 10 days, respectively [50]. Thus, eggs that experience more than ca. 2 weeks at similar temperatures in the field would likely exhibit significant mortality, hindering long-term establishment. The moderate cold stress limit corresponded with areas where the daily average *T_min_* fell below −16 °C for ca. 9 consecutive days. We further calibrated the moderate cold stress limit to reduce the exclusion of records during years in which high levels of cold stress accumulated (2001 and 2005) (Appendix A).

After the cold stress limit calibrations, five presence records for the SLF were excluded by severe or moderate cold stress across multiple (2–11) modeled years. The average weekly *T_min_* values at locations for both records were ca. −20 °C, which has been estimated as the pest’s lower lethal temperature [50]. However, we chose to keep the final parameters because the records lacked precise geographic coordinates and were from a single study [63]. Additionally, predictions of exclusion by cold stress in most areas north of ca. 40° N for multiple modeled years are consistent with surveys indicating that this latitude appears to be the northern range limit for this pest in China [64].

#### 2.4.2. Heat Stress Parameters

An analysis of the estimates of *T_max_* of the hottest week for each presence record for the SLF from China indicated that 99% (389/392) of the records occurred in areas with an average weekly *T_max_* of ≤37 °C (Appendix A). The three remaining records were from parts of Zhejiang Province, where up to four consecutive days above 37 °C occurred during the year with the most heat stress accumulation (2008). These findings are consistent with laboratory studies that found high or complete mortality of nymphs after exposure to ca. 3 days at 35–40 °C [24,38] and a significant decline in egg survival at temperatures > 40 °C [65]. We therefore calibrated the moderate heat stress limit to correspond roughly to areas that experienced a daily average *T_max_* of >37 °C for more than five consecutive days and to minimize the exclusion of records across all the modeled years. The severe heat stress limit corresponded with areas that experienced a daily average *T_max_* of >37 °C for ca. 15 consecutive days.

#### 2.4.3. Climatic Suitability Model Validation

Predictions of the potential distribution of the SLF for 20 recent years were combined for each region (North America and China) to identify areas that were consistently predicted to be suitable. We estimated cold and heat stress accumulations for North America, based on 20-year climate averages and extreme years in terms of stress accumulations, to provide insight into their potential role in limiting the SLF’s distribution. To validate the climatic suitability model, we assessed whether DDRP correctly included presence records from the United States in the potential distribution for the SLF for each of the 20 modeled years.

We compared the potential distribution of the SLF for North America to a map depicting county-level detection records for the tree of heaven (*Ailanthus altissima* (Mill.) Swingle) and American Viticultural Areas [66,67] to identify areas where suitable climates for the SLF co-occurred with its preferred host plant and with wine-grape-growing regions, respectively. Presence records for the tree of heaven were derived from EDDMapS (N = 18,112) [60] and GBIF (N = 40,128) [68]. GBIF records with a geographic uncertainty of >10 km were removed. A single record was retained for each county, resulting in 1539 records from 49 U.S. states and 11 Canadian provinces and territories.

### 2.5. Model Demonstration

We produced phenological event maps of first egg hatch and first adult emergence for North America and Europe for 2023 to demonstrate the DDRP model and provide insight into the SLF’s phenology and potential distribution during a year that experienced record-breaking warm temperatures (NOAA website https://www.ncei.noaa.gov/access/monitoring/monthly-report/global/202313, accessed on 18 March 2025). The SLF has not yet established in Europe; however, much of the region is at risk owing to suitable climates and host plant availability [13,14]. Climate data for Europe were derived from the E-OBS database at a spatial resolution of 0.1° (ca. 11.1 km^2^; https://surfobs.climate.copernicus.eu, accessed on 28 August 2024) [69].

## 3. Results

### 3.1. Phenology

According to the analyses of monitoring data for the SLF, the DOY_pred_ for the first appearance of adults exhibited the tightest fit with DOY_obs_, with the lowest error rate (MAE = 4.2) and the shortest prediction interval (average length = 22 days) (Table 3, Figure 2). The MAE and prediction intervals were higher and longer for the other three phenological events; however, the sample sizes were very small (N = 5). The DOY_pred_ for the first egg hatch exhibited the lowest correspondence with DOY_obs_, with the longest prediction interval (average length = 151 days) and the highest error rate (MAE = 28) and bias (−27.8 days).

In the analysis that used the iNaturalist data, DDRP predicted the first appearances of early nymphs, late nymphs, adults, and first-generation eggs significantly earlier than the DOY_obs_ values of these life stages (Table 4, Figure 3 and Figure 4). Additionally, the DOY_pred_ of the last egg hatch occurred significantly later than the DOY_obs_ of the overwintered eggs. Most observations for the United States (66–99%) had life stages that corresponded to the one predicted by the model (Figure 3, Appendix A). Observations of younger life stages (e.g., a late nymph was observed but an adult was predicted) ranged from 21 to 49%, which may indicate model underprediction. Conversely, observations of older life stages than those predicted by the model ranged from 0 to 3.2%, indicating a very low incidence of model overprediction. The results for China indicated a somewhat greater extent of both under- and overprediction, depending on the observed life stage; however, the sample sizes were very small (N < 10) (Figure 4).

### 3.2. Climatic Suitability

The climatic suitability model for the SLF correctly included >99.9% (14,986/14,987) of the presence records in the potential distribution for North America (Figure 5B), which provides evidence for a high degree of model sensitivity. The potential distribution, according to predictions for 20 recent years, was consistent with the pest’s documented range in China (Figure 5) [25,64]. In North America, it overlapped with all the counties where the tree of heaven occurs and with all the American Viticulture Areas, except for the Central Valley of California for particularly hot years. Conversely, areas excluded from the potential distribution across all the years were predominantly in the coldest areas of Canada, including Ontario and Quebec, as well as very hot areas of the American Southwest (e.g., the Sonoran and Mohave Deserts). For some modeled years, climate stresses excluded the SLF from higher-latitude areas of midwestern and northeastern states, as well as parts of the Central Valley of California and Texas.

Cold stresses played a larger role in shaping the SLF’s potential distribution in North America than heat stresses (Figure 6). On average, across 20 years, severe cold stress was predicted to exclude the SLF from most of eastern Canada and northern parts of North Dakota and Minnesota in the United States (Figure 6A). Areas excluded by moderate cold stress extended into the upper Midwest and Northeast, which suggests that long-term establishment may be difficult in these regions. Heat stresses excluded the SLF only from hot parts of the desert Southwest, southwestern Texas, and parts of Mexico (Figure 5B). Areas excluded by moderate and severe heat stresses expanded throughout this area and into the southern United States (Texas and Oklahoma) during an extreme year in terms of heat stress accumulation (2023).

### 3.3. Predictions for North America and Europe for 2023

Predictions of the first egg hatch and the first appearance of adults for the SLF for North America and Europe for 2023 varied substantially by latitude. For northern regions, the first egg hatch occurred mostly between May and July (Figure 7A and Figure 8A), whereas the first adults appeared between July and September (Figure 7B and Figure 8B). The earliest dates for these events were in southern regions of both continents, with egg hatch occurring as early as January in areas of the Gulf Coast. There was insufficient degree-day accumulation for adults to appear in areas north of ca. 48.5° N in North America and ca. 54.8° N in Europe, as well as in high-elevation areas of both continents (e.g., the Cascades and Rocky Mountains). This finding may suggest that the SLF would be unable to complete its life cycle in these areas.

For North America, cold stresses for 2023 excluded the SLF from most of eastern Canada except for the Atlantic coastline and southern Ontario and Quebec, whereas heat stresses excluded the pest from hot areas of Mexico, the American Southwest, and Texas (Figure 7). For Europe, cold stresses excluded the pest from high-latitude areas of Northern Europe, whereas heat stresses excluded it only from small parts of southern Spain and Turkey (Figure 8).

## 4. Discussion

Early detection and rapid response programs are integral to minimize the spread of the SLF to new regions, such as the western United States and Europe, where the pest is expected to cause major disruptions to fruit, ornamental, and forest industries [3,6,70]. We presented a spatialized model of phenology and climatic suitability for the SLF for use in the DDRP platform, which serves as an open-source decision support tool to help to detect, monitor, and manage invasive threats [33]. The near-real-time forecasts produced by the model are available at USPest.org (https://uspest.org/CAPS, accessed on 29 July 2025) and at the USA National Phenology Network (https://usanpn.org/data/maps/forecasts/spotted_lanternfly, accessed on 29 July 2025) to support decision making for the CONUS. Forecasts of the egg hatch and the appearance of adults are particularly relevant for surveillance to prevent new establishments and for managing existing populations.

### 4.1. Phenology Model

The validation analyses that used monitoring data from the United States revealed varying levels of concordance between predicted and observed dates of phenological events. Predictions for the first appearance of adults were the most precise (average prediction interval length = 22 days) and had the lowest error rate (MAE = 4.2 days). Error rates and prediction intervals were higher and longer, respectively, for the first egg hatch, nymphs halfway developed, and the first oviposition. However, very small sample sizes for each event (N = 5) may lead to greater uncertainty in parameter estimates and less reliable results. Daily monitoring data from a larger number of locations across the SLF’s range are needed to robustly evaluate the model. Data collected for state and federal monitoring programs are typically less useful for model validation because they lack precise survey locations (e.g., only county names are provided) and because traps are only checked weekly or seasonally.

The validation analyses that used the iNaturalist data indicated a very low incidence of model overprediction (≤3.2%) for nymphs, adults, and first-generation eggs, which provides evidence for good model performance. Observations that occurred earlier than predicted dates may indicate model error; however, observer error may also play a role. For example, location errors may occur when data are uploaded from a place other than the location where the species was originally observed [71]. The latest date of egg hatch was predicted after overwintered eggs were observed for >99% of the observations, which provides additional evidence that the model predicts the correct life stage. Our study adds to an increasing number of studies that use iNaturalist data for developing and validating phenology models [72,73]. Nevertheless, a primary limitation of these data is that they represent encounters with an organism on a single date, which provides no information on when the observed life stage first appeared or when it finished development. Thus, we could not assess the precise extent of model over- vs. underpredictions using iNaturalist data.

Geographic variation in the dates when SLF adults appear will likely translate into geographic differences in host-plant pressure. Adults are more damaging than nymphs because they can feed on larger branches or trunks with more sap flow, which results in more honeydew production and faster depletion of host resources [5]. According to DDRP predictions for 2023, the SLF reached the adult stage ca. 6 months earlier in Florida than in the northernmost parts of its potential distribution in North America (March vs. October). Host plants in the southern CONUS may experience greater impacts by the SLF because adults would emerge earlier in the year and experience killing frosts later in the year, resulting in a longer feeding period compared to those in northern regions. In Europe, the pest may exert the highest feeding pressure on host plants in the Mediterranean Basin, where adults are predicted to appear ca. 5 months earlier than those in northern Europe (June vs. October). The SLF is univoltine, even in warmer regions of China, but its potential to have more than one generation per year is not well understood [23,43].

The cohort parameter settings in DDRP produced a distribution in the hatch times of overwintered eggs that corresponded well with the field data used for the model calibration; however, the distributions likely vary across the SLF’s geographic range. In a study of field-collected eggs that were moved to a 25 °C chamber, the eggs that experienced overwintering periods of longer than 5 months had a narrower duration of hatch and earlier initial hatch dates [44]. Thus, the distribution of the egg hatch times in DDRP may need to be narrower in cold regions, with an earlier low bound for the minimum degree-days to complete egg development (xdist1). Modifying the phenology model to accept a raster of values for cohort numbers and cohort parameter values is one approach to accommodate geographic variation in overwintered eggs’ development.

Including a mechanism in DDRP for predicting diapause termination, based on the fulfilment of chilling requirements, could be another potential approach to account for variation in egg hatch times in the SLF. However, this would require more data on factors that influence diapause in the SLF, including the role of temperature in diapause initiation and termination and other potential cues, such as photoperiod [24,43]. The pest’s diapause requirements are highly variable, and the roles of fluctuating temperatures and temperatures of <5 °C in the three stages of egg development are not fully understood [43]. Evidence for within- and among-population variations in the development of nymphs (lower developmental threshold and stage duration) [24] may suggest that accommodating for developmental variation in nymphs may also improve the model’s realism.

DDRP predicts the potential distribution of the SLF using survival climate stresses; however, the establishment risk may also be influenced by the pest’s ability to complete its life cycle—i.e., to reach the adult stage and lay eggs. Most parts of Canada and northern Europe had insufficient degree-day accumulation for adults to appear for 2023, which may suggest that the establishment risk is lower for these areas, even in the absence of cold stress. Similarly, a model of the development and survival of the SLF based on 30-year climate averages (1970–2000) indicated that temperatures were too cold for the adult stage to complete in northern Europe [14]. Overlaying DDRP predictions produced for multiple years would provide additional insights into spatiotemporal variability in the establishment risk based on life cycle completion. With additional data on the effect of temperature on egg diapause in the SLF, future modeling work could assess whether an ability to terminate diapause in warm regions may hinder establishment.

Host plant attributes that may affect the SLF’s development and survival may be a source of error in the DDRP model. In particular, host plant species may significantly affect the development time for nymphs, with temperature and humidity potentially interacting with the effects of host plant species on nymphal development [58]. The SLF exhibits higher rates of survival and development when it has access to the tree of heaven, its preferred host [74,75,76,77,78]. Development is also affected by the variety and age of grape vines (*Vitis* spp.) [76,78,79], host plant health, and nutritional quality of host tissues [80,81]. Accounting for potential host impacts on SLF development in the DDRP model is possible; however, knowing which model to use for different areas would be difficult because of the co-occurrence of different host species. Additionally, the pest exhibits differences in host utilization throughout the season as well as within life stages [7,54].

### 4.2. Climatic Suitability Model

The climatic suitability model for the SLF had a very high degree of sensitivity (>99.9% of the presence records were correctly modeled), which provided evidence that the SLF’s potential distribution for North America was not underpredicted. Estimating the model specificity (i.e., the proportion of absence records correctly modeled) and overall model accuracy will require observations of where the SLF is known to be excluded by the climate. The potential distribution across most of the or all 20 modeled years (2004–2023) overlapped with regions with susceptible plant production industries, including all the American Viticulture Areas and states with substantial productions of fruit trees, hops (*Humulus* spp.), and grapes (*Vitis* spp.) (e.g., California, Washington, and New York) [11]. This finding is consistent with a model that identified these regions as being at high risk of wine market disruption by the SLF, based on the pest’s transport, climate, and impact potentials [18]. As the SLF is expected to heavily impact the wine and grape industries [18,70], this finding suggests that the pest should be closely monitored in these regions.

DDRP’s exclusion of the coldest regions of the CONUS (i.e., the Midwest and Northeast) and most of eastern Canada across most of or all the modeled years is mostly consistent with Maxent models [16,17] and a CLIMEX model [12] for the SLF, all of which predicted low suitability or unsuitable conditions in these areas. However, no area of the CONUS was excluded across all 20 years, which suggests that the SLF may be capable of short-term establishment, even in the coldest areas, such as northern Minnesota. The SLF is not known to be established in Canada but has been sighted across several provinces, where it threatens the wine and grape industries [82].

DDRP excluded only the hottest areas of western North America from the SLF’s potential distribution, whereas previous climatic suitability models predicted unsuitable conditions across most of this region [12,16,17]. This discrepancy is likely due to the previous models’ use of moisture variables and their reliance on presence records from China for model fitting (Maxent) [16,17] or for calibrating moisture parameters (CLIMEX) [12]. The SLF’s documented distribution in China excludes xeric regions, but it is unclear whether this is due to aridity or an absence of the tree of heaven, which occurs predominantly in northeast and central China [64]. DDRP’s inclusion of areas with suitable temperatures for the SLF in the arid West encompassed regions where the tree of heaven is invasive, such as in riparian zones and irrigated areas, and where vulnerable plant hosts, such as grape and fruit trees, are grown. Thus, these areas may be at higher risks of establishment than predicted by previous climatic suitability models.

Most regions of Europe were included in the potential distribution for the SLF for 2023, which is consistent with previously published climatic suitability models for this pest, based on historical climate averages [12,13,16]. The continent is also considered as being at high risk owing to the widespread availability of host plant species for the SLF, including the tree of heaven and various species of grapes, fruit trees, woody trees, and ornamentals [13]. The establishment of the SLF in Europe may be particularly catastrophic for grape-growing countries (e.g., France, Spain, Italy, Portugal, and Romania), which produced over half of all global wine exports and contributed over EUR 34 billion in value in 2021 [83].

Potential sources of error in the climatic suitability model include the assumptions that the SLF’s tolerance to climate stress is constant within and across populations and that temperature is the only survival-limiting climatic factor. The SLF may exhibit higher survival rates in cold temperatures when it experiences alternating temperature regimes [24]. However, DDRP is not programmed to allow for recovery from thermal stresses or to attain higher resistance to stresses under certain conditions. Levine et al. [84] reported a positive effect of the body size on the tolerance of SLF females to hot temperatures, as well as a tendency for females to be larger in urban areas. Uncertainty in thermal tolerances for the SLF could be communicated by running multiple versions of the model, each with different values for certain parameters (e.g., the cold or heat stress thresholds or stress limits). Precipitation is known to decrease egg viability in the SLF [85], which suggests that the addition of precipitation factors to the model could potentially improve the performance for areas with large amounts of winter rain or snowfall.

### 4.3. Near-Real-Time Forecasts for Decision Support

The near-real-time forecasts for the SLF, at USPest.org (https://uspest.org/CAPS, accessed on 29 July 2025), include all the predictions produced by DDRP in both gridded (GeoTIFF) and image (PNG) formats. The USA National Phenology Network (https://usanpn.org/data/maps/forecasts/spotted_lanternfly, accessed on 29 July 2025) presents the predictions of the first egg hatch and the appearance of adults as Pheno Forecasts [86] in both static and interactive formats, and end users can sign up to receive e-mail notifications that provide advance warnings (three, two, and one week(s) before) of when these events will occur in their area. Forecasts of egg hatch provide insight into when nymphs will appear, which may help to detect the SLF before it can spread to additional locations. Additionally, they may help with the timing of spray applications that target the egg stage, which are more effective when applied approximately two weeks before hatch [28]. Forecasts for adults may help to time treatments that are applied to control adults on infested trees, which may also reduce the spread of the pest.

Near-real-time forecasts could have applications for SLF management using biological control agents, which are currently under investigation and not yet available for widespread release. Forecasts of egg hatch may provide better insight into when to release the nymphal parasitoid, *Dryinus sinicus* Olmi, 1987 (Hymenoptera: Dryinidae), which is most effective against the first instars of the SLF [87]. Additionally, forecasts of oviposition may help to identify areas where egg availability will align with the emergence of egg parasitoids, such as *Anastatus orientalis* Yang and Choi, 2015 (Hymenoptera: Eupelmidae) or *Ooencyrtus kuvanae* (Howard, 1910) (Hymenoptera: Encyrtidae) [88]. Forecasts of the appearance of adults may be useful for timing the release of entomopathogenic fungi that tend to infect adults at higher levels than nymphs, such as *Beauveria bassiana* and *Batkoa major* [26,27].

Operationalizing DDRP for real-time decision support for regions beyond the CONUS where the SLF is invasive (e.g., Japan and South Korea [25,89]) or at high risk of establishment (e.g., Canada, Europe, and Australia) could be challenging owing to delays in climate data releases. For example, Daymet data for North America are released ca. 4 months after the current calendar year, which hinders their use for real-time modeling for the entire continent. Capinha et al. (2024) [72] produced short-term forecasts of the phenology for several invasive species in Europe, using gridded climate data from the National Oceanic and Atmospheric Administration’s Global Forecast System (GFS) (https://www.ncei.noaa.gov) [90]. However, the coarse spatial resolution of the GFS data (0.25° = ca. 27.75 km^2^) may limit the usefulness of forecasts for small or topographically complex areas.

### 4.4. Other Applications

Future work could explore integrating DDRP forecasts of the SLF with predictions of its dispersal [11,91,92,93,94] to provide a more comprehensive assessment of both where and when to expect this pest. The rapid spread of the SLF is driven in part by human-mediated dispersal, in which SLF egg masses and adults latch onto vehicles, such as cars, trucks, and trains [93,95]. Surveillance could focus on areas that lack survival-limiting climate stresses, have sufficiently warm temperatures to complete the life cycle, and are at high risk of being colonized in the next several years owing to their proximity to human transportation systems or suitable host plants. In the areas at the highest risk of establishment, pest managers could use forecasts of egg hatch to ensure the timely surveillance of nymphs, which are easier to detect than the rather cryptic egg masses.

Future modeling in DDRP could explore how the SLF’s distribution and phenology may shift owing to contemporary and future climate change [37]. Trend analyses of predictions for multiple years may reveal potential shifts in the dates of management-relevant events in the SLF, such as egg hatch, or in survival-limiting climate stresses [37]. Increases in winter temperatures due to global warming are predicted to reduce egg mortality in the SLF, facilitating its range expansion [22]. Consistent with this expectation, Maxent models indicate the potential for a northward expansion of the SLF in future climate-change scenarios [13,17]. Areas with the tree of heaven that occur north of the SLF’s current distribution may be at the highest risk of establishment under future climate conditions [64].

A detailed analysis of DDRP predictions and phenological observations may provide insight into whether the SLF’s phenology has evolved in the United States. A previous analysis of iNaturalist data for the SLF documented a shift to earlier activity and a lengthening of the life cycle (earlier emergence and later winter die off) as the pest expanded its range between 2014 and 2022 [96]. However, it was unclear whether evolutionary change during the invasion, phenotypic plasticity in different environments, or other unmeasured factors may explain these trends [96]. Our model was developed using data from some of the oldest known populations in the United States, so the differences in the model-predicted vs. observed dates of events may be larger in recently invaded areas if the SLF is evolving to have earlier phenological events.

## 5. Conclusions

We developed a DDRP model that integrates the mapping of the phenology and climatic suitability for the SLF to provide insight into when to expect important events in this pest in areas that are at risk of establishment. Overall, the model exhibited a good performance, but additional monitoring data collected from across the SLF’s range are needed to further validate and potentially improve upon the model. Cold and heat stresses may be insufficient to prevent the long-term establishment of the SLF in most parts of the CONUS, but an inability for the pest to complete its life cycle could potentially reduce the establishment risk for some areas. The near-real-time forecasts produced by the model are available at two websites to support decision making.

## Figures and Tables

**Figure 1 insects-16-00790-f001:**
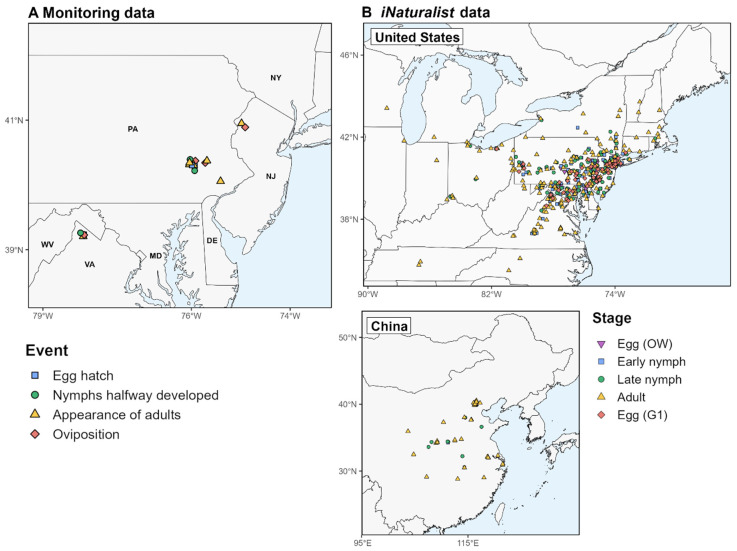
Geographic origins of observations of the spotted lanternfly (SLF) used to validate the phenology model. (**A**) Observations from monitoring studies represent the first dates of phenological events at locations in the eastern United States. (**B**) Observations in iNaturalist represent single encounters with the SLF at a specific time and location in the United States and China (top and bottom maps, respectively). A single location per county (United States) is shown to reduce clutter.

**Figure 2 insects-16-00790-f002:**
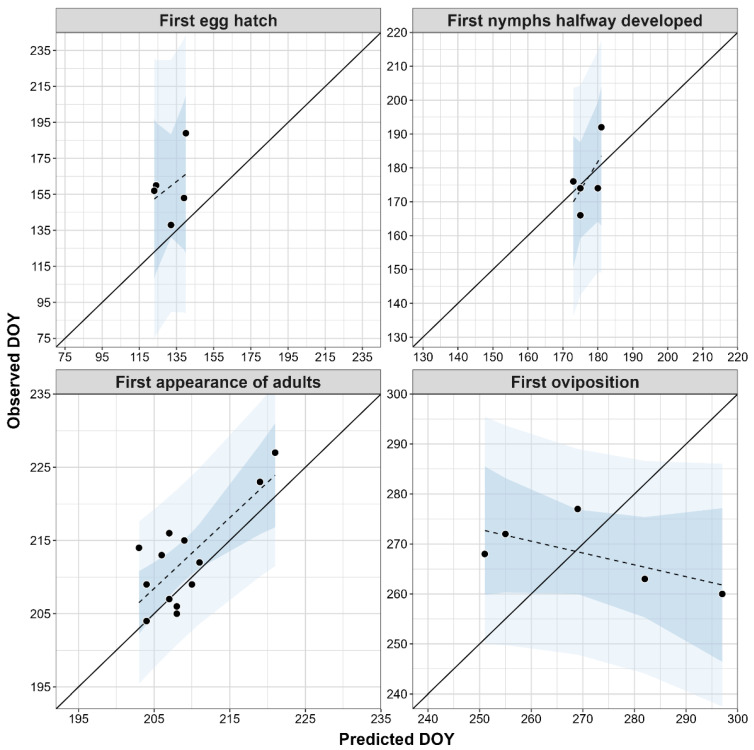
Scatterplots of predicted vs. observed dates (the day of the year, DOY) for four phenological events of the spotted lanternfly, based on published monitoring data for the United States. Each plot depicts the line of best fit calculated with a linear regression (dashed line), a 1:1 relationship between field- and model-predicted DOYs (black line), a 95% confidence interval (darker-blue shading), and a 95% prediction interval (light-blue shading).

**Figure 3 insects-16-00790-f003:**
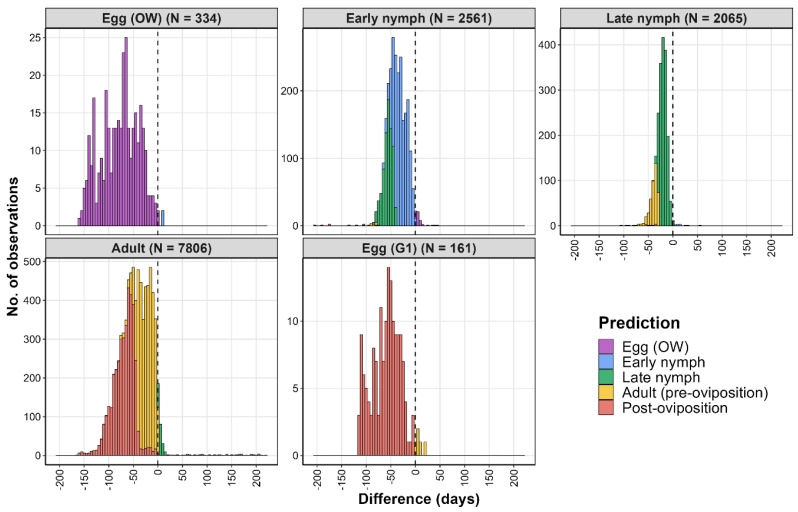
Histograms showing differences between predicted vs. observed dates (days) of life stages of the spotted lanternfly in the United States, based on the iNaturalist data. A value of 0 indicates no difference between dates (dashed line). Positive values may indicate model overprediction, wherein a life stage was observed before it was predicted to be present. For overwintered (OW) eggs, positive values indicate that egg hatch was predicted before eggs were observed. G1 = first generation.

**Figure 4 insects-16-00790-f004:**
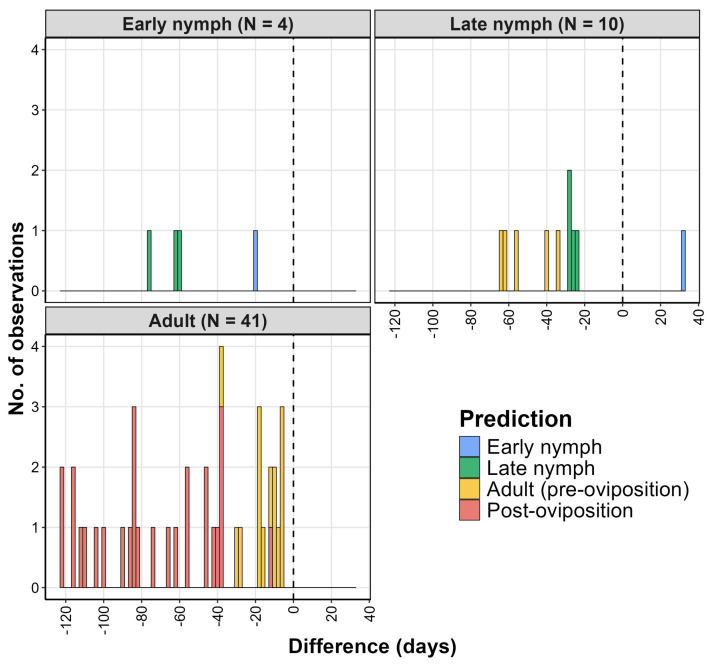
Histograms showing differences between predicted vs. observed dates (days) of life stages of the spotted lanternfly in China, based on the iNaturalist data. A value of 0 indicates no difference between dates (dashed line). Positive values may indicate model overprediction, wherein a life stage was observed before it was predicted to be present.

**Figure 5 insects-16-00790-f005:**
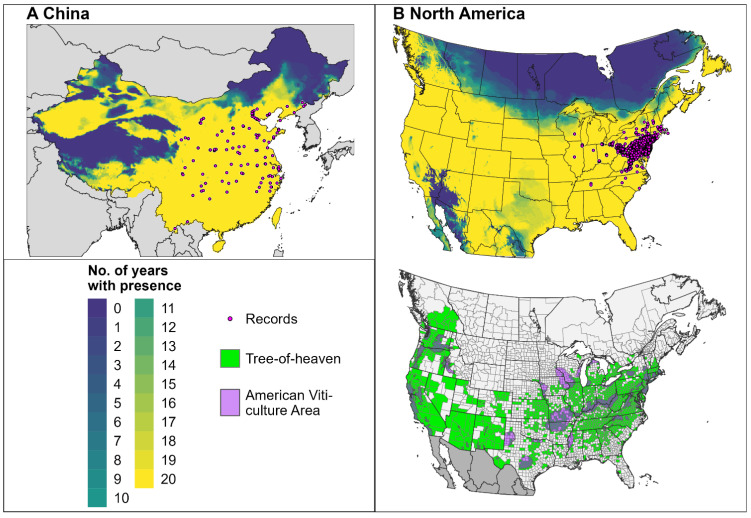
The modeled potential distributions for the spotted lanternfly in (**A**) China and (**B**) North America (top maps) according to DDRP runs for 20 recent years. Yellow areas were included in the potential distributions for all 20 years, whereas areas with cooler colors were excluded by climate stress for one or more years. Pink circles depict the approximate locations of the presence records used for model calibration and validation. The bottom map in (**B**) depicts county-level detections of the tree of heaven for Canada and the United States, as well as American Viticulture Areas.

**Figure 6 insects-16-00790-f006:**
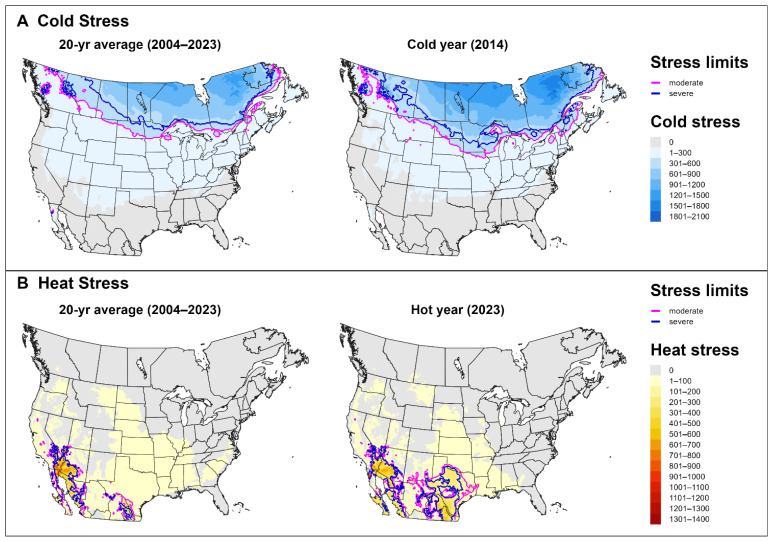
Maps of annual (**A**) cold stress and (**B**) heat stress accumulations for the spotted lanternfly in North America, produced by DDRP. Averages of stress accumulations over a 20-year period (2004–2023) (left panels) are compared to accumulations for an extreme year (right panels) in terms of the total area excluded by the cold or heat stress (2014 and 2023, respectively).

**Figure 7 insects-16-00790-f007:**
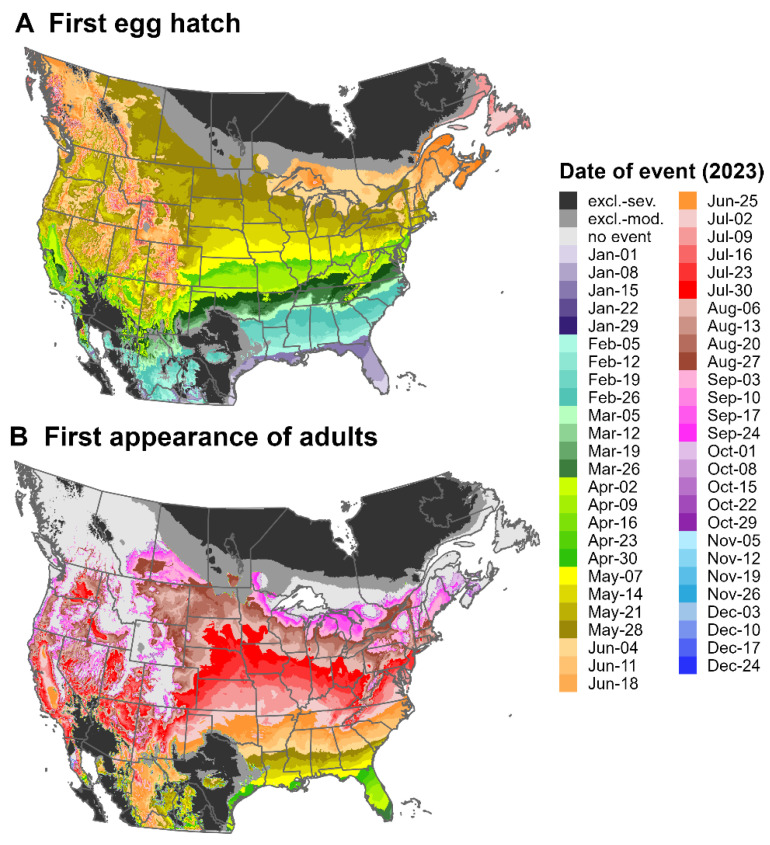
Maps of the predicted dates of (**A**) the first egg hatch and (**B**) the first appearance of adults for the spotted lanternfly for North America for 2023, produced by DDRP. The maps include estimates of climatic suitability, where long-term establishment is indicated by areas not under moderate (“excl.-moderate”, medium gray) or severe (“excl.-severe”, dark gray) climate stress exclusion. Areas with insufficient degree-day accumulation for an event to occur are indicated with light gray.

**Figure 8 insects-16-00790-f008:**
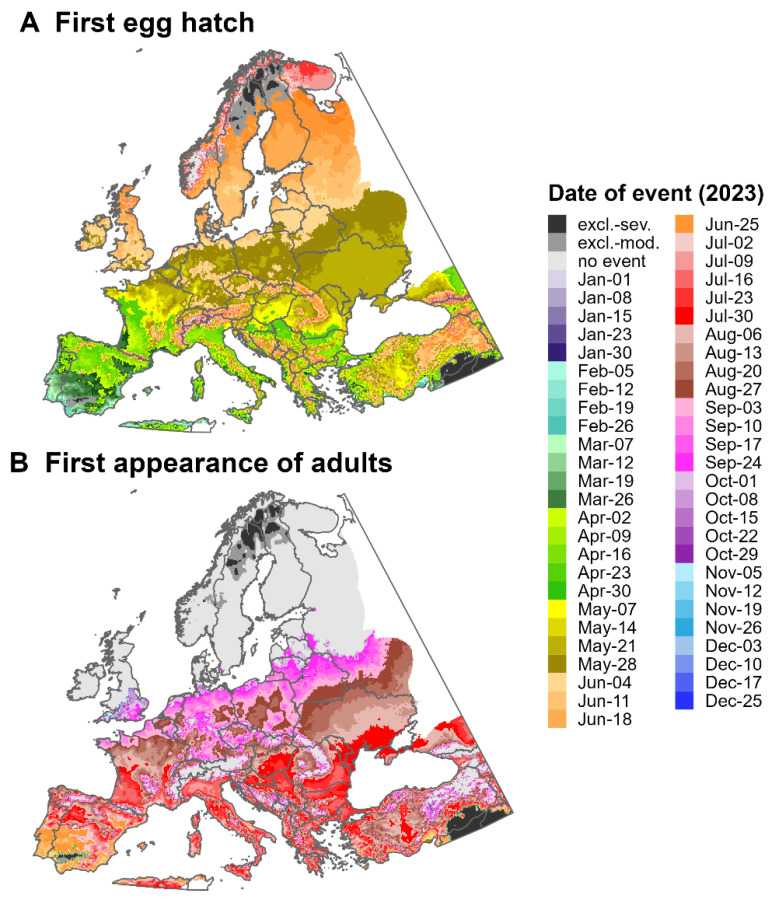
Maps of the predicted dates of (**A**) the first egg hatch and (**B**) the first appearance of adults for the spotted lanternfly for Europe for 2023, produced by DDRP. The maps include estimates of climatic suitability, where long-term establishment is indicated by areas not under moderate (“excl.-moderate”, medium gray) or severe (“excl.-severe”, dark gray) climate stress exclusion. Areas with insufficient degree-day accumulation for an event to occur and missing climate data are indicated with light gray and white, respectively.

**Table 1 insects-16-00790-t001:** Parameters used in the DDRP model for the SLF.

Parameter	Code	Value
Lower developmental thresholds (°C)		
Eggs	eggLDT	10
First- and second-instar nymphs	larvaeLDT	10
Third- and fourth-instar nymphs	pupaeLDT	10
Adults	adultLDT	10
Upper developmental thresholds (°C)		
Eggs	eggUDT	35
Larvae	larvaeUDT	35
Pupae	pupaeUDT	35
Adults	adultUDT	35
Stage durations (°C degree-days)		
Egg	eggDD	202
Larval (L1–L4)	larvaeDD	890
Pupal (pre-oviposition adult)	pupDD	630
Adult (50% oviposition)	adultDD	146
Phenological events (°C degree-days)		
Egg hatch	OWEventDD	varies
Nymphs halfway developed	larvaeEventDD	442
Appearance of adults	pupaeEventDD	1
Oviposition	adultEventDD	1
Diapausing eggs	eggEventDD	100
Cold stress		
Cold stress temperature threshold (°C)	coldstress_threshold	−16
Cold stress degree-day (°C) limit when most individuals die	coldstress_units_max1	300
Cold stress degree-day (°C) limit when all the individuals die	coldstress_units_max2	475
Heat stress		
Heat stress temperature threshold (°C)	heatstress_threshold	37
Heat stress degree-day (°C) limit when most individuals die	heatstress_units_max1	115
Heat stress degree-day (°C) limit when all the individuals die	heatstress_units_max2	175
Cohorts		
Degree-days (°C) to complete egg development (average)	distro_mean	190
Degree-days (°C) to complete egg development (variation)	distro_var	15,000
Minimum degree-days (°C) to complete egg development	xdist1	135
Maximum degree-days (°C) to complete egg development	xdist2	360
Shape of the distribution	distro_shape	normal
Other		
Order of stages	stgorder	OE, L, P, A, E
Obligate diapause (1 = TRUE)	obligate_diapause	1
Degree-day calculation method	calctype	triangle

LDT = lower developmental threshold, UDT = upper developmental threshold, DD = degree-day, L1–L4 = larval instars 1–4, OW = overwintered, OE = overwintered egg, L = larval, P = pupal, A = adult, E = egg.

**Table 2 insects-16-00790-t002:** Summary of iNaturalist data used to validate the phenology model for the spotted lanternfly. The country of origin, number of observations for each life stage (N_obs_), and number of years (N_years_) for which observations were available are indicated. All the observations were derived from unique sites.

Country	Life Stage	N_obs_	N_years_
United States	Egg (OW)	334	6
United States	Early nymph	2561	9
United States	Late nymph	2065	7
United States	Adult	7806	9
United States	Egg (G1)	161	7
China	Early nymph	4	4
China	Late nymph	10	6
China	Adult	41	10

OW = overwintered, G1 = first generation.

**Table 3 insects-16-00790-t003:** Summary of the results of the model validation analyses that used monitoring data for the spotted lanternfly. The observed phenological event, number of observations (N_obs_), mean absolute error (MAE), bias with standard deviation (SD), range of differences between predicted and observed dates (in days), and average length of the prediction interval (Length) are shown for each event. MAE = the average absolute difference of (observed − predicted), Bias = average of (predicted − observed), and SD = standard deviation of (predicted − observed).

Modeled Event	Observed Event	N_obs_	MAE	Bias	SD	Range	Length
First egg hatch	First nymphs	5	28	−28	17	−49 to −6	151
First nymphs halfway developed	First appearance of third instars	5	6	0	8	−11 to 9	65
First appearance of adults	First appearance of adults	13	4	−3	4	−11 to 3	22
First oviposition	First oviposition	5	20	3	24	−17 to 37	44

**Table 4 insects-16-00790-t004:** Summary of the results produced by the model validation analyses for the spotted lanternfly that used the iNaturalist data for the United States and China. The total number of observations for each life stage (N_obs_) and the median of the predicted vs. observed dates of the life stage (DOY_pred_ and DOY_obs_, respectively) are indicated. The percentages (%) of the observations in which DOY_pred_ < DOY_obs_ are indicated for the early nymph, late nymph, adult, and first-generation (G1) egg stages. For the overwintered (OW) egg stage, the percentages of the observations in which DOY_pred_ > DOY_obs_ are shown because the first-predicted event in the model is egg hatch, which would occur after eggs are observed. Asterisks indicate statistical significance (paired Wilcoxon signed-rank test, one-tailed).

Region	Observed Event	N_obs_	DOY_pred_	DOY_obs_	% Observations
United States	Egg (OW)	334	151 ***	78	99.4
United States	Early nymph	2561	126 ***	165	98.2
United States	Late nymph	2065	176 ***	198	99.4
United States	Adult	7806	206 ***	255	96.9
United States	Egg (G1)	161	250 ***	308	93.1
China	Early nymph	4	110 *	159	100
China	Late nymph	10	156 **	189	90.0
China	Adult	41	178 ***	230	100

* *p* < 0.1; ** *p* < 0.05; *** *p* < 0.01.

## Data Availability

The DDRP code and species parameter file used for this study were archived in Zenodo (https://doi.org/10.5281/zenodo.15538509). The most recent code for DDRP is available at GitHub (https://github.com/bbarker505/ddrp_v3.git, accessed on 26 July 2025).

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
