# Peer review of "Real-Time Integrative Mapping of the Phenology and Climatic Suitability for the Spotted Lanternfly, Lycorma delicatula"

_insects, 2025, doi:10.3390/insects16080790_

Round 1

Reviewer 1 Report

Comments and Suggestions for Authors

Using day-degree model and climatic suitability can effectively map potential infestation area of invasive insect pests. However, I have some suggestions/comments that may improve the manuscript.

Keywords: "Lycorma delicatula" is not necessary as it is in the title already. "development" => "development model"

1. Introduction: The following could be added to help explain why linear day-degree model is used and improve the model performance. 1) the advantage and disadvantage of linear day-degree model vs non-linear model (eg logistic) for simulating thermal accumulation of insect development; 2) the diapause mechanism of SLF eggs: entry condition (thermal accumulation before winter), termination requirements (chilling effect), and the percentage of post-diapause development (for better estimating hatching date); 3) the likely spread mechanisms: short-distance expansion (hop and fly) and long-distance distribution (passive transportation, such as its introduction into PA).

"...on temperatures that were cooler than present-day climates..." because the trend of global warming?

"However, none of these models simulate developmental variation within populations..." we always assume population structure has a normal distribution. So the timing of "first sight", "sudden increase" and "peak size" is used for different purposes.

2. Methods: 

2.1 Climate data

If no terrain elevation from a digital elevation model, do we need the fine resolution of 4 km2? As 0.1 deg is about 10 km while 4 km2 grid is 2 km resolution, having not reduced too much (only doubled).

2.2 Phenology model

2.2.1 Life cycle and overwintering stage

"The model uses a start date of 1 January..." do we know by this point the egg diapause is terminated or not (that is the chilling effect is fulfilled)? And what the percentage of embryo development when it enters diapause? If this is true, then we can start to summarize thermal accumulation from this point for the rest of embryo development and thus forecast hatching date. Otherwise, we should run the model from September when earliest eggs could be laid and estimate the date when the embryos go into diapause (and also measuring the chilling effect from then).

2.2.2 Thresholds, stage durations, and phenological events

Although the linear degree-day model is easy, it won't address the slow development at lower temperatures (eg post-diapause embryo development). Surely it doesn't perform well at high temperatures either, but for SLF, egg and nymph development is in spring and early summer and thus we may not need to consider the slow-development at high temperatures.

"At a lower threshold of 10 deg, egg, nymphal..." these values are in both Tab S1 and Tab 1, and thus no need to be listed here. However, these values may need to estimate if the model starts from the autumn (September for example from egg-laying).

We may need to explain how to calculate daily average temperatures from daily maximum and minimum temperatures, eg two sine waves with the consideration of sun rise and sunset?

2.2.3 Phenology model validation

"Egg observations were not included because of uncertainty as to whether they belonged to the overwintered vs first generation." this is not true. If eggs before September, most likely the last batch of overwintering eggs. Otherwise, they would be the earliest batch of newly laid eggs.

Tab 4 & Fig 3: predictions are all earlier than observations for both nymph and adult, possible slow-development rates at high temperatures (i.e. heat stress)?

4.2 Climate suitability model

"DDRP's exclusion of the hottest areas..." should we consider SLF egg require a period of chilling effect to break diapause?

Hope this helps.

Author Response

Comment 1: Using day-degree model and climatic suitability can effectively map potential infestation area of invasive insect pests. However, I have some suggestions/comments that may improve the manuscript.

Response 1: We appreciate the questions and suggestions for making our spotted lanternfly (SLF) model more mechanistic and potentially more accurate. However, the ideas exceed our current goals and objectives in creating a practical, real-time decision support model that is both achievable and useful. If our goal had been to create a more research-oriented model and more data on SLF’s overwintering biology were available, then the suggestions would be more relevant to our study. We would not in that case, have succeeded in implementing the model, as it is now deployed on our servers with daily updates. We have already seen evidence of adoption of our SLF model for decision support, though we have not yet documented the degree of adoption at this early stage.

Comment 2: Keywords: "Lycorma delicatula" is not necessary as it is in the title already. "development" => "development model"

Response 2: We made this correction.

Comment 3: 1. Introduction: The following could be added to help explain why linear day-degree model is used and improve the model performance. 1) the advantage and disadvantage of linear day-degree model vs non-linear model (eg logistic) for simulating thermal accumulation of insect development; 2) the diapause mechanism of SLF eggs: entry condition (thermal accumulation before winter), termination requirements (chilling effect), and the percentage of post-diapause development (for better estimating hatching date); 3) the likely spread mechanisms: short-distance expansion (hop and fly) and long-distance distribution (passive transportation, such as its introduction into PA).

Response 3: The trade-offs between linear vs. non-linear models for insect development is a long-standing discussion that should not be re-hashed in an applied decision support study such as this paper. We have, however, added a new section to the Methods (Section 2.1. Modeling overview) that justifies our use of the linear model:

“DDRP assumes long established norms for phenological modeling used in decision support, such as making use of linear degree-day models using daily maximum and minimum temperatures (Tmin and Tmax, respectively) as model inputs. Details on the mechanics and data requirements of DDRP already exist [33,37] and are, therefore, only summarized for this study. We believe that the more advanced use of non-linear equations is not warranted in the case of this model because (1) temperature-development studies and data used to parameterize SLF development rates [30,38] were well-fitted using the linear approach (see Section 2.3.2); (2) a non-linear modeling approach may not offer significant improvement in model accuracy in general [39] and would require temporal resolutions (hourly or better) that are not currently available from high quality spatial data sources; and (3) spatial models such as DDRP would be challenged by computational limits if non-linear developmental submodels and higher resolution temperature data were to be incorporated. Therefore, we adhered to the simpler, widely used standards in the use of linear degree-days in the DDRP platform.”

Comment 4: "...on temperatures that were cooler than present-day climates..." because the trend of global warming?

Response 4: Yes. We have edited this sentence to read “…were cooler than present-day climates owing to global warming.”

Comment 5: "However, none of these models simulate developmental variation within populations..." we always assume population structure has a normal distribution. So the timing of "first sight", "sudden increase" and "peak size" is used for different purposes.

Response 5: We’re not entirely sure how to address this comment. Our point here is that previous phenology models for SLF assume that a single population at any given site develops in synchrony – i.e., there is no developmental variation. Thus, an event like first adult emergence would occur on a single date at any given location. In contrast, DDRP models include developmental variation by the use of cohorts, in which cohorts retain an overall normal distribution that can be expressed as percentages in transitions and stages, e.g. first, 5%, peak, and 50% emergence. With 7 cohorts, there will be 7 dates of emergence at a location, with the first cohort exhibiting the earliest date of first emergence.

Comment 6: 2. Methods: 

2.1 Climate data

If no terrain elevation from a digital elevation model, do we need the fine resolution of 4 km2? As 0.1 deg is about 10 km while 4 km2 grid is 2 km resolution, having not reduced too much (only doubled).

Response 6: Nearly all gridded climate data we are aware of, including the formats used in this study (Daymet and CDAT), do in fact incorporate effects of elevation (and other topographical effects) on climate. It is therefore worthwhile to run models at the highest possible spatial resolution, such as 4km2. We have access to higher resolution 800m data (PRISM) but have not found its usage to be justifiable based on modeling at the extent of 48-state conterminous U.S., for example.

Comment 7: 2.2 Phenology model

2.2.1 Life cycle and overwintering stage

"The model uses a start date of 1 January..." do we know by this point the egg diapause is terminated or not (that is the chilling effect is fulfilled)? And what the percentage of embryo development when it enters diapause? If this is true, then we can start to summarize thermal accumulation from this point for the rest of embryo development and thus forecast hatching date. Otherwise, we should run the model from September when earliest eggs could be laid and estimate the date when the embryos go into diapause (and also measuring the chilling effect from then).

Response 7: The DDRP platform lacks the overwintering mechanism proposed here and none of our 18 models include chilling requirement parameters. We therefore assume that SLF responds to temperatures over 10oC, which does not tend to happen where SLF is currently distributed in the U.S. until the spring. We are satisfied that this assumption is apt in this case. As demonstrated in the Methods (Section 2.3.2., third paragraph), the cohort parameter settings in DDRP produced a distribution in hatch times of overwintered eggs that corresponded well with field data used for model calibration. Additionally, our new analysis of overwintered egg observations from the iNaturalist dataset demonstrated that >99% of eggs were observed prior to the predicted date of last egg hatch.

Modifying DDRP to include the proposed overwintering method would be laborious and may not be justified until more data on SLF’s diapause requirements are collected. Keena and Nielsen (2021) studied how chilling periods affected rates of egg hatch and found that egg development preceded faster at 25oC if eggs were incubated at temperatures <= 10oC. However, they concluded that more research studies are needed on the complicated effects of temperature on the three phases of egg diapause to fully predict when egg hatch will occur for all climatic regimes to which the insect may be exposed. Additionally, they reported that the effects of fluctuating temperatures and temperatures < 5oC on the three stages of egg development are not understood, and that diapause requirements are highly variable. We have added additional content to the Discussion (Section 4.1, fourth paragraph) about potential methods and data sources that future modeling work could incorporate to potentially improve predictions of hatch times in SLF.

References

Keena, M.A., and A. L. Nielsen. 2021. Comparison of the hatch of newly laid Lycorma delicatula (Hemiptera: Fulgoridae) eggs from the United States after exposure to different temperatures and durations of low temperature. Environ. Entomol. 50: 410–417

Comment 8: 2.2.2 Thresholds, stage durations, and phenological events

Although the linear degree-day model is easy, it won't address the slow development at lower temperatures (eg post-diapause embryo development). Surely it doesn't perform well at high temperatures either, but for SLF, egg and nymph development is in spring and early summer and thus we may not need to consider the slow-development at high temperatures.

Response 8: As noted in our response to the previous comment, there are insufficient data on SLF’s development at lower temperatures to develop a robust non-linear model. Furthermore, we are not aware of any studies in which non-linear development of insects at the low end of the temperature-development trend have been verified in the field. Our assumption has been that any such errors at lower temperatures would not accumulate to a measurable degree, and that using field data to calibrate springtime activities will mitigate such potential errors. As for hotter temperatures, we parameterize upper developmental thresholds (35oC in this case) whereby development no longer increases with temperature. This approach should be reasonable for a mobile species such as SLF which (except for the egg stage) is able to move to cooler, shaded environments when temperatures are hot.

Comment 9: "At a lower threshold of 10 deg, egg, nymphal..." these values are in both Tab S1 and Tab 1, and thus no need to be listed here. However, these values may need to estimate if the model starts from the autumn (September for example from egg-laying).

Response 9: Most of the information in Table S1 was indeed redundant with Table 1 so we removed it. We have responded to the suggestion of an autumn start date above.

Comment 10: We may need to explain how to calculate daily average temperatures from daily maximum and minimum temperatures, eg two sine waves with the consideration of sun rise and sunset?

Response 10: We’re not sure if this comment refers to the estimation of hourly temperatures for use in a non-linear temperature-devleopment model or to the method that we used to calculate degree-days in the DDRP model. As stated above, the calculation of heat units on an hourly basis would require hourly gridded data which is largely unavailable (except in near-term forecasts), and would overly encumber the computational load on our servers. We utilized the single triangle method of calculation of daily degree-days from daily Tmax and Tmin (noted in Table 1 and second paragraph of Section 2.3.2). This long-standing method is more accurate than the method of using daily average temperatures for heat unit accumulations. Additionaly, it has been shown to be very close in accuracy to the Baskerville-Emin (1969) sine curve method (Wilson and Barnett 1983), which has been the standard recommended by U.C. Davis IPM for many decades now (see https://ipm.ucanr.edu/WEATHER/ddconcepts.html). We therefore use the single triangle method as the standard for our spatial models due to more efficient computing speed.

References

Baskerville, G.L., and P. Emin. 1969. Rapid estimation of heat accumulation from maximum and minimum temperatures. Ecology. 50: 514-517.

Wilson, L.T., and W.W. Barnett. 1983. Degree-days: an aid in crop and pest management. California Agriculture. 37:4-7.

Comment 11: 2.2.3 Phenology model validation

"Egg observations were not included because of uncertainty as to whether they belonged to the overwintered vs first generation." this is not true. If eggs before September, most likely the last batch of overwintering eggs. Otherwise, they would be the earliest batch of newly laid eggs.

Response 11: We have now included egg observations in iNaturalist data for SLF in model validation analyses. Previously, the source of uncertainty was egg observations that were collected in July and August. However, we realized upon inspecting photos for these observations that the eggs had hatched (i.e., they had finished development). These observations were excluded from validation analyses (please see the Methods for more details).

Comment 12: Tab 4 & Fig 3: predictions are all earlier than observations for both nymph and adult, possible slow-development rates at high temperatures (i.e. heat stress)?

Response 12: We only used iNaturalist observations to assess whether nymphs and adults were predicted earlier than observed because their date of first appearance is unknown. For example, an adult could have been present for several days at a site before someone noticed it and entered the observation into the database. Thus, we expected predicted dates to be earlier than observed dates.  Additionally, we usually build slightly conservative models, because early predictions tend to be more helpful and less costly than late predictions. As no models are perfect, we prefer to err more on the side of early than late to be most useful. We have added some additional content to clarify methods that used iNaturalist data (Section 2.3.3, second paragraph) as well as the limitations of these data (Section 4.1, second paragraph). 

Comment 13: 4.2 Climate suitability model

"DDRP's exclusion of the hottest areas..." should we consider SLF egg require a period of chilling effect to break diapause?

Response 13: Climate suitability modeling in DDRP is separate from the phenology model and involves estimating and accumulating survival-limiting temperature stresses to predict the potential distribution of a species (we re-worded the first sentence of Section 2.4 of the Methods to clarify this point). Thus, the addition of diapause into the phenology model would not affect whether SLF’s is excluded by heat stress. However, we agree that an inability to break diapause owing to insufficient egg chilling could potentially limit SLF’s distribution in particularly warm regions and have added this point to the Discussion.

Reviewer 2 Report

Comments and Suggestions for Authors

The article presents the results of modeling the phenology of a dangerous invasive pest: the spotted lanternfly.
The problem statement and methodology are clear. The results obtained are convincing and the conclusions drawn from them are discussed in detail.
These conclusions have important practical significance for predicting the spread and harmfulness of the spotted lanternfly in a changing climate.
I am involved in taxonomy and faunistics. Therefore, my comments in text relate to the spelling of Latin names of species.

Author Response

Comment 1: The article presents the results of modeling the phenology of a dangerous invasive pest: the spotted lanternfly. The problem statement and methodology are clear. The results obtained are convincing and the conclusions drawn from them are discussed in detail. These conclusions have important practical significance for predicting the spread and harmfulness of the spotted lanternfly in a changing climate.

Response 1: Thank you for reviewing our manuscript and for the positive feedback!

Comment 2: I am involved in taxonomy and faunistics. Therefore, my comments in text relate to the spelling of Latin names of species.

Response 2: We have made the suggested corrections to species names in the manuscript. However, we did not delete the scientific name in the first line of the Introduction Some readers may skip reading the Simple Summary and Abstract and therefore not see the full species name. Additionally, our review of recently published papers in the Insects journal all mention the full species names in the Introduction even in cases where it is also noted in the Abstract and Simple Summary.

Comment 3: Abstract: Do authors need to mention climate change over the last 20 years when the pest appeared in the US in 2014?

Response 3: We deleted “across 20 recent years” from the Abstract to minimize confusion. As described in the Methods (Section 2.3), we analyzed climate data for China and North America for 20 recent years to assess whether DDRP correctly included areas where SLF has been observed in the potential distribution. Thus, we were not assessing the impacts of climate change on SLF in the U.S.

The attached PDF contains point-by-point responses to each of the comments and requested edits.
